# Orbital Neurolymphomatosis in Patient with CNS Lymphoma

**DOI:** 10.3390/diagnostics15060780

**Published:** 2025-03-20

**Authors:** Tara Shooshani, Michael Han, Jeremiah P. Tao, Samuel J. Spiegel, Maria Del Valle Estopinal

**Affiliations:** 1School of Medicine, University of California, Irvine, CA 92697, USA; 2Department of Ophthalmology, University of California, Irvine, CA 92697, USAsjspiege@hs.uci.edu (S.J.S.); 3Department of Pathology, Division of Ophthalmic Pathology, University of California, Irvine, CA 92868, USA

**Keywords:** neurolymphomatosis, CNS lymphoma, histopathology

## Abstract

Neurolymphomatosis (NL) is a rare manifestation of hematologic malignancies, characterized by a neoplastic infiltration of the peripheral nervous system and cranial nerves (CNs). Non-Hodgkin lymphomas (NHLs) account for 90% of NL cases, while acute leukemia represents 10% of the cases. NL can occur as the first manifestation of a malignancy (primary), or as a relapse or progression of a previously treated disease (secondary). Herein, we report a unique case of NL involving the left orbit and CNs in a 74-year-old female with primary central nervous system (CNS) diffuse large B-cell lymphoma (DLBCL). Our patient developed secondary neurolymphomatosis involving the left orbit and CNs II, III, V, and VI, supported by clinical, radiologic, and histologic findings. The lacrimal gland enhancement was histopathologically proven to be caused by the direct spread of CNS DLBCL to the lacrimal nerve, a branch of CN V, identifying NL as one of the conditions that can affect this organ. The lacrimal gland could be considered as a more accessible biopsy site when the involvement of CN V is suspected.

**Figure 1 diagnostics-15-00780-f001:**
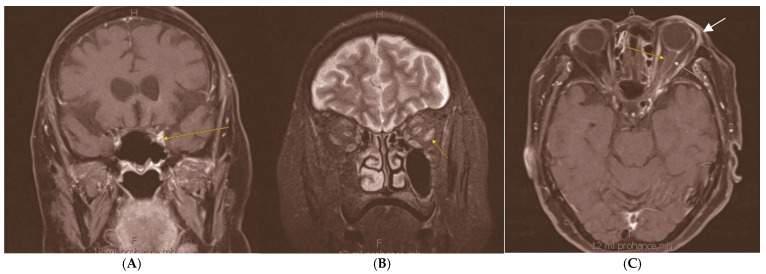
Magnetic resonance imaging (MRI) of the brain and orbits with and without contrast demonstrated enhancement of left orbital apex and optic nerve sheath, extraocular muscles, cavernous sinus, and superficial lacrimal gland. (**A**) demonstrates a fat suppression technique with short TI inversion recovery (STIR), disclosing enhancement of left orbital apex (arrow). (**B**,**C**) show T1 post-contrast axial imaging of the orbit with fat suppression, with (**B**) depicting enhancement of left lateral rectus muscle (arrow) and (**C**) highlighting enhancement of optic nerve sheath (narrow yellow and white arrows) and superficial lacrimal gland (solid white arrow). The patient presented with newly developed progressive headaches, diplopia, and ptosis. Her medical history was relevant for mixed-mechanism glaucoma, proliferative diabetic retinopathy, and CNS DLBCL, which were actively being treated with systemic therapy encompassing methotrexate, temozolomide, rituximab, ibrutinib, and dexamethasone. The left eye demonstrated visual acuity of 20/50 OS (20/40 OD), mildly decreased color vision, and a relative afferent pupillary defect. Sensorimotor exam revealed ophthalmoplegia in all directions, with profound limitations in abduction and elevation, as well as ptosis of the left eyelid. The dilated fundoscopic exam showed prior panretinal photocoagulation and mild left optic nerve pallor. The cerebrospinal fluid (CSF) cytology was negative for lymphoma, which flow cytometry analysis (FCM) supported. No infectious agents were detected in CSF, blood, or urine. (H—head, F—feet, A—anterior, P—posterior).

**Figure 2 diagnostics-15-00780-f002:**
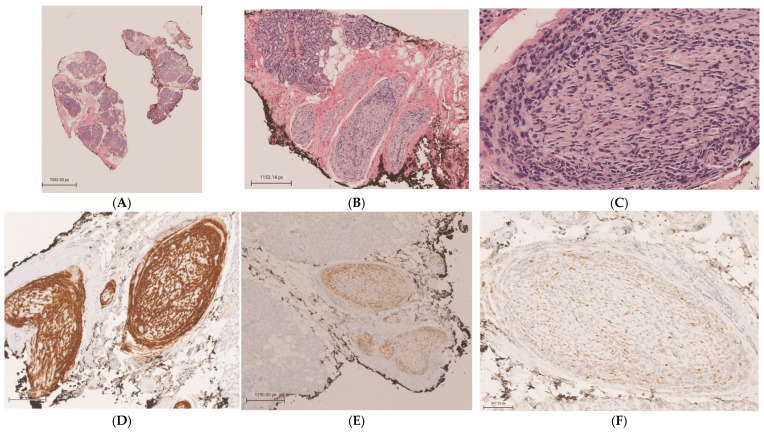
A biopsy of the left lacrimal gland was obtained for histopathologic assessment and FCM. Low power microscopic examination revealed lobules of mature lacrimal gland acini and ductal structures against a background of stromal fibrosis (H&E, 1×) (**A**). Nerves within the stroma were infiltrated and expanded by atypical round blue cells (H&E, 5.31×) (**B**). Atypical cells infiltrate the endoneurium of a nerve, mixed with apoptotic bodies and mitotic figures (H&E, 28×) (**C**). Mixed B and T lymphocytes (9%) with high nonspecific binding and no diagnostic immunophenotype were detected by FCM. Immunohistochemical studies were performed in a Clinical Laboratory Improvement Amendment of 1988 (CLIA 88) certified laboratory, encompassing the following findings: cluster of differentiation 20 (CD20) diffusely expressed on the surface of neoplastic large B lymphocytes, highlighting the B-cell lineage of the malignant cells (4×) (**D**); paired box 5 (PAX-5) showing positive nuclear expression in tumor cells, consistent with B-cell phenotype (4×) (**E**); weakly positive expression of cellular myelocytomatosis oncogene (c-Myc) in 20–30% of lymphoma cells, indicating variable oncogene activation (14.4×) (**F**); and high Ki-67 proliferation index in more than 70% of tumor cells (4×) (**G**). Scattered small T lymphocytes percolating nerves and acinar structures were positive for cluster of differentiation 3 (CD3) (6×) (**H**). Anti-cytokeratin cocktail (AE1/AE3) was negative in neoplastic cells (4×) (**I**). The immunophenotype of the malignant lymphocytes was similar to the previously described in the frontal lobe biopsy which supported the diagnosis of secondary neurolymphomatosis involving lacrimal gland nerves. NL is a rare disease and challenging to diagnose, with an estimated prevalence of 0.2% in NHL [1]. Men are affected more commonly than women in the 6th to 7th decades [1,2]. Painful neuropathy or polyradiculopathy, painless polyneuropathy, cranial neuropathy, and peripheral mononeuropathy are some of the clinical manifestations [1]. However, these findings can be elusive, mimicking non-neoplastic and paraneoplastic neuropathies. The malignant nature of NL lymphoid cells distinguishes it from the benign infiltrates of lymphocytes that characterize paraneoplastic or inflammatory neuropathies and the peripheral neuropathy induced by cancer treatments. A combination of clinical suspicion and imaging are required to identify NL. While MRI usually demonstrates nonspecific abnormal findings, positron emission tomography–computer tomography (PET-CT) is highly sensitive (87–100%) and helps to localize an optimal biopsy site in the peripheral nerves [2]. In addition, the use of CSF cytopathologic evaluation and flow cytometry analysis increases the diagnostic sensitivity and specificity. The biopsy of the involved structure remains the gold standard, despite limitations due to the location of the target structure and the potential morbidity to neurologic function. Ocular manifestations of NL are uncommon. Palsies of CNs III and VI, orbital apex syndrome, trigeminal neuropathy, and secondary involvement of ciliary nerves, choroid, and optic nerves have been described in the literature [3,4,5]. Fritzhand et al. [4] reviewed 18 cases of neurolymphomatosis involving cranial nerves near the orbit and found that its incidence was more associated with solid tumors in the orbit than with infiltrative processes or hematologic spread. CN V2 and the cavernous sinus were the most common sites of spread. To our knowledge, only three previous case reports confirmed, histologically, the ocular involvement by NL, and the diagnosis was determined by autopsy in two of these cases [3,4]. Treatment requires chemotherapy alone or combined with radiotherapy. NL’s median survival has been reported as 10 months, with a 36-month survival proportion of 24% [2]. Recently, the use of rituximab has improved the progression-free survival [6,7]. Notably, our case demonstrated lacrimal gland nerve involvement by the CD20-positive B lymphoma cells despite ongoing systemic therapy including rituximab (monoclonal antibody targeting CD20 B-cell surface antigen), which suggests poor response to the treatment and progression of the disease. The patient’s diplopia improved mildly after increasing the dexamethasone dosing. Unfortunately, two months later, the patient succumbed to diffuse cerebral vasculitis and lymphoma. In conclusion, the prompt and accurate diagnosis of NL is necessary for a better clinical outcome. We recommend considering NL in the differential diagnosis of lacrimal gland disorders.

## Data Availability

No new data were created or analyzed in this study. Data sharing is not applicable to this article.

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
