# Peer review of "Orbital Neurolymphomatosis in Patient with CNS Lymphoma"

_diagnostics, 2025, doi:10.3390/diagnostics15060780_

Round 1
Reviewer 1 Report
Comments and Suggestions for Authors
Thanks for this interesting image. Some comments:
1-Line12. This sentence "Lymphomas account for 90% of the cases, while 0.2% occur in Non- 12
Hodgkin Lymphoma.". Lymphomas vs what? NHL? HL? In my experience the majority of cases are NHL. I imagine you refer to 0.2% of NHL presenting with NL. Please clarify.
2-In line 28 clarify if Rituxan was administered intrathecally and if you are referring to retention of CD20 in extrameningeal biopsies.
Comments on the Quality of English LanguagePlease clarify the sentence in comment 1.
Author Response
1-Line12. This sentence "Lymphomas account for 90% of the cases, while 0.2% occur in Non- 12
Hodgkin Lymphoma.".
Response: thank you for your review. I made some edits on the statement for clarification. Please see lines 12, 13 and 60.
2-In line 28 clarify if Rituxan was administered intrathecally and if you are referring to retention of CD20 in extrameningeal biopsies.
Response: the patient received systemic therapy. Yes, the biopsy of the lacrimal gland revealed CD20-positive B lymphoma cells infiltrating nerves, despite ongoing systemic therapy with rituximab, suggesting poor response to the treatment and progression of the disease. I made some edits on those statements and added reference #7 for clarification. Please see lines 78-80.
Reviewer 2 Report
Comments and Suggestions for Authors
This short communication submitted for Interesting Images concerns a case of neurolymphomatosis (NL) involving the orbit and the cranial nerves II, III, V, and VI, in a patient previously diagnosed with primary central nervous system (CNS) diffuse large B-cell lymphoma (DLBCL). It is highlighted the fact that lacrimal gland biopsy, followed by histopathologic examination and immunohistochemistry, was able to prove the infiltration of the lacrimal nerve with the same lymphomatous B cells found at diagnosis of DLBCL. Both MRI and histopathologic images are representative for this case and clearly interpreted.
There are some minor issues to be addressed:
- Abstract: The statement “Lymphomas account for 90% of the cases, while 0.2% occur in Non-Hodgkin Lymphoma” has no meaning. It should be reformulated as such: „Non-Hodgkin lymphomas account for 90%, and acute leukemias for the 10% of cases”.
- Discussion: The statement “Noteworthy, in our case the tumor cells retain strong CD20 expression, suggesting nonresponse to rituximab (monoclonal antibody targeting CD20 B-cell surface antigen), and corroborating her poor clinical outcome” is also confusing. If the neoplastic cells express CD20 strongly why it is assumed a lack of response to rituximab?
Author Response
- Abstract: The statement “Lymphomas account for 90% of the cases, while 0.2% occur in Non-Hodgkin Lymphoma” has no meaning. It should be reformulated as such: „Non-Hodgkin lymphomas account for 90%, and acute leukemias for the 10% of cases”. Response: thank you for your review. I made some edits on the statement for clarification. Please see lines 12, 13 and 60.
- Discussion: The statement “Noteworthy, in our case the tumor cells retain strong CD20 expression, suggesting nonresponse to rituximab (monoclonal antibody targeting CD20 B-cell surface antigen), and corroborating her poor clinical outcome” is also confusing. If the neoplastic cells express CD20 strongly why it is assumed a lack of response to rituximab? Response: I made some edits on those statements and added reference #7, for clarification. The lacrimal gland nerve involvement by CD20-positive B lymphoma cells despite ongoing systemic therapy with rituximab suggests poor response to the treatment and progression of the disease. Please see lines 78-80. Thank you for your review.